# Advancing Cancer Treatment by Targeting Glutamine Metabolism—A Roadmap

**DOI:** 10.3390/cancers14030553

**Published:** 2022-01-22

**Authors:** Anna Halama, Karsten Suhre

**Affiliations:** Department of Physiology and Biophysics, Weill Cornell Medicine-Qatar, Doha 24144, Qatar

**Keywords:** cancer, metabolism, glutamine metabolism, cancer treatment, glutaminolysis inhibition, drug resistance

## Abstract

**Simple Summary:**

Dysregulated glutamine metabolism is one of the metabolic features evident in cancer cells when compared to normal cells. Cancer cells utilize glutamine for energy generation as well as the synthesis of other molecules that are critical for cancer growth and progression. Therefore, drugs targeting glutamine metabolism have been extensively investigated. However, inhibition of glutamine metabolism in cancer cells results in the activation of other metabolic pathways enabling cancer cells to survive. In this review, we summarize and discuss the targets in glutamine metabolism, which has been probed in the development of anticancer drugs in preclinical and clinical studies. We further discuss pathways activated in response to glutamine metabolism inhibition, enabling cancer cells to survive the challenge. Finally, we put into perspective combined treatment strategies targeting glutamine metabolism along with other pathways as potential treatment options.

**Abstract:**

Tumor growth and metastasis strongly depend on adapted cell metabolism. Cancer cells adjust their metabolic program to their specific energy needs and in response to an often challenging tumor microenvironment. Glutamine metabolism is one of the metabolic pathways that can be successfully targeted in cancer treatment. The dependence of many hematological and solid tumors on glutamine is associated with mitochondrial glutaminase (GLS) activity that enables channeling of glutamine into the tricarboxylic acid (TCA) cycle, generation of ATP and NADPH, and regulation of glutathione homeostasis and reactive oxygen species (ROS). Small molecules that target glutamine metabolism through inhibition of GLS therefore simultaneously limit energy availability and increase oxidative stress. However, some cancers can reprogram their metabolism to evade this metabolic trap. Therefore, the effectiveness of treatment strategies that rely solely on glutamine inhibition is limited. In this review, we discuss the metabolic and molecular pathways that are linked to dysregulated glutamine metabolism in multiple cancer types. We further summarize and review current clinical trials of glutaminolysis inhibition in cancer patients. Finally, we put into perspective strategies that deploy a combined treatment targeting glutamine metabolism along with other molecular or metabolic pathways and discuss their potential for clinical applications.

## 1. Introduction

The molecular landscape of cancer cells strongly differs from that of normal cells and contributes to features that are known as cancer hallmarks [1]. The dysregulated metabolism, adjusted to meet the increased energy and biomass production demands, enabling sustained cell division, was described as one of the vital cancer hallmarks [1]. There is a wide spectrum of metabolic processes activated by cancer cells [2]. These extend beyond the initial observation of increased glucose consumption with simultaneous lactate production, independent of the oxygen availability, known as the Warburg effect [3]. For instance, accelerated purine and pyrimidine metabolism and enhanced lipid synthesis are features significantly contributing to the proliferative capacity of cancer cells [4,5]. To cope with this demand, cancer cells modulate their nutrient acquisition mode and redirect their metabolic pathways [2]. The enhanced fluxes of glucose, various amino acids (e.g., glutamine, arginine, serine, and branched-chain amino acids (BCAAs)), folate, and unsaturated fatty acids were previously described along with altered tricarboxylic acid (TCA) cycle metabolism [2,5,6,7,8,9]. Yet, dysregulated glutamine (Gln) metabolism was recognized as central to cancer cell fitness due to its contribution to various metabolic pathways including synthesis of nucleotides, lipids, nonessential amino acids (NEAAs), energy generation, and redox homeostasis.

The metabolic alterations attributed to cancer cells have become an attractive treatment target. One of the first treatment strategies utilizing dysregulated cancer metabolism was targeting elevated nucleotide synthesis [10], proposed by Sidney Farber, the father of modern chemotherapy [11].

Identification of Gln metabolism as central to cancer cells resulted in multiple attempts probing it as a treatment target. Molecules predominantly hampering glutaminolysis by inhibiting glutaminase (GLS), the enzyme catalyzing metabolism of Gln to glutamate, were shown to attenuate cancer growth and proliferation. One of the GLS inhibitors CB-839 (telaglenastat) is currently being investigated in multiple different clinical trials, frequently in combination with chemotherapeutics. Although inhibition of GLS reduces cancer proliferation, we and others have shown that such interventions result in activation of alternative metabolic compensations enabling cancer cell survival under the treatment [12,13]. Therefore, to design rational treatment strategies efficiently targeting cancer, it is critical to carefully analyze those compensatory pathways.

In this review, we focus on various aspects of dysregulated glutamine metabolism observed in cancer cells and its interaction with molecular signaling. We further summarize the treatment strategies proposed to target dysregulated Gln metabolism and will comment on pathways activated in response to such a treatment as potential cancer survival options. Finally, we put in perspective combined treatment options targeting glutamine metabolism along with other molecular or metabolic pathways and discuss their potential for clinical applications. The interplay between Gln metabolism and the immune system, along with Gln metabolism targets contributing to enhanced cancer immunotherapy, was recently reviewed [14] and will not be covered in this review.

## 2. Overview of Glutamine Metabolism

The amino acid glutamine (Gln) can be synthesized by the human body and is thus considered as a nonessential amino acid (NEAA) [15]. Nevertheless, due to the involvement of Gln in multiple processes, under specific conditions, this amino acid was recognized as conditionally essential [16]. Among all the amino acids, Gln is the most abundant in the body [17] and its concentration depends on the overall body metabolism; the balance between exogenous supply, endogenous synthesis, and release; and its utilization by various tissues. The Gln metabolism across different organs was captured for the first time by Adolf Krebs in 1935 [15]. The Gln synthesis preliminary occurs in the skeletal muscle, lungs, and adipose tissue and is orchestrated by glutamate ammonia ligase (GLUL), also known as glutamine synthase (GS), an enzyme that resides in the cytosol and that catalyzes Gln synthesis from glutamate and ammonium ions (NH^4+^). The catabolism of Gln is preliminary conducted by the intestinal mucosa, immune cells, renal tubule cells, and liver, where glutaminase (GLS), located in mitochondria, catalyzes the conversion of Gln into glutamate [18].

### 2.1. Organ-Dependent Glutamine Fate

The role of Gln depends on the tissue of origin. For instance, Gln synthesized in the muscle is released into the bloodstream to support visceral organs with amino nitrogen. Corticosteroids stimulate Gln synthesis and its release by the skeletal muscle, which can be seen during stress conditions such as fasting, injury, and illness when increased Gln amount is released [19]. The primary site for the utilization of released Gln is the gastrointestinal tract and the liver. The cells of the gastrointestinal tract, e.g., colonocytes and enterocytes, utilize Gln as the main respiratory fuel, rather than glucose [20]. The nitrogen generated from Gln during this process is used to synthesize ammonia, alanine, proline, and citrulline, which are released into the bloodstream and then utilized by the liver and kidney [21]. In the liver, the absorbed Gln supports the proliferation of hepatocytes and generation of energy, as well as contributing to gluconeogenesis (*de novo* glucose synthesis) [15,22]. Additionally, the liver utilizes nitrogen from Gln to fuel the urea cycle and urea formation, which serves as detoxification and regulation of the blood pH [23]. In the kidney, Gln-dependent ammoniagenesis enables the generation of bicarbonate and thus maintains the acid–base balance in the body [24]. The Gln absorbed by the kidney is also utilized for gluconeogenesis and contributes to blood glucose homeostasis [25]. Importantly, the cells of the immune system, including lymphocytes, macrophages, and neutrophils, depend more on Gln than on glucose [26,27,28]. Immunological challenges increase the demand for Gln in those cells [29]. Immune cells utilize Gln both for energy generation and synthesis of macromolecules [30]. In summary, Gln metabolism is regulated in an organ-dependent manner and contributes to a broad spectrum of processes in the body.

### 2.2. Glutamine Metabolism on the Cellular Level

Extracellular Gln cannot cross the plasma membrane and requires transporters to enter the cells. There are in total 14 such transporters, classified into four different families: SLC1, SLC6, SLC7, and SLC38, which can all support either Gln influx into the cell or efflux from the cell into the extracellular space [31]. These transporters differ in their substrate specificity (as most of the transporters contribute to the transport of other neutral or cationic amino acids), ion- and pH-dependence, and role under physiological and cancer conditions, as described in great detail by Bhutia et al. [31].

Once in the cell, Gln can be incorporated into a versatile spectrum of metabolic pathways in which Gln-derived nitrogen or carbon are utilized. As a nitrogen source, Gln is particularly used for the synthesis of nucleotides (pyrimidines and purines), NEAAs, and glucosamine, whereas carbon from Gln is used for gluconeogenesis, TCA cycle, and glutathione metabolism.

*De novo* nucleotide synthesis, critical for DNA and RNA, is supported by Gln, which contributes along with nitrogen to the assembly of purines and pyrimidines [32]. In the purine synthesis pathway, Gln serves as a substrate for three enzymes: (1) phosphoribosyl pyrophosphate amidotransferase (PPAT), involved in the conversion of 5-phosphoribosyl-1-pyrophosphate (PRPP) into 5-phosphoribosyl-1-amine (PRA) by deploying nitrogen from Gln [33]; (2) 5-phosphoribosylformylglycinamidine synthase (FGAM synthase), which catalyzes the formation of formylglycinamidine ribonucleotide in an ATP-dependent reaction in which the amino group of glutamine is transferred into formylglycinamidine ribonucleotide (FGAR) [34]; and (3) guanosine monophosphate (GMP) synthetase, which catalyzes the formation of GMP in an ATP-dependent reaction in which amino group from glutamine is transferred to replace the oxygen in the C-2 position of xanthine monophosphate (XMP) [35]. In the pyrimidine pathway, Gln serves as a substrate in two reactions: (1) ATP-dependent synthesis of carbamoyl phosphate, the first step in pyrimidine synthesis, catalyzed by the enzyme carbamoyl phosphate synthetase II (CPS II) [36], and (2) synthesis of cytidine triphosphate (CTP) by amination of uridine triphosphate (UTP), catalyzed by CTP synthase in an ATP-dependent reaction [37].

Gluconeogenesis is a critical process for maintaining glucose homeostasis in the body under different conditions, including fasting [38]. It was shown that Gln is a major gluconeogenic precursor [39]. The process of glucose formation from Gln requires two deamination steps; the first is catalyzed by GLS and results in the formation of glutamate [15], and the second is catalyzed by glutamate dehydrogenase (GDH) and leads to the synthesis of α-ketoglutarate [40]. There are two GLS isoforms, one characterized as kidney (also known as brain)-type (GLS1) and the other defined as liver-type (GLS2) [15,41]. The acquired α-ketoglutarate is incorporated into the TCA cycle where it is metabolized into malate. The malate can be either further oxidized in the mitochondria by malate dehydrogenase (MDH) into oxaloacetate or transported into the cytoplasm and converted into oxaloacetate by the cytosolic form of MDH [42]. The obtained oxaloacetate is decarboxylated to phosphoenolpyruvate (PEP) by the mitochondrial or cytosolic form of phosphoenolpyruvate carboxykinase (PEPCK), dependent on the reaction site [43]. The PEP, synthesized in mitochondria, is transported into the cytosol, and the cytosolic PEP is used for gluconeogenesis [44].

Alternatively, under conditions where glucose is not required, PEP can be metabolized by pyruvate kinase (PK) into pyruvate [45], which further can be incorporated into the lipogenic pathway.

In fact, Gln can serve as a source of lipid synthesis via two routes: (1) the oxidative route, involving metabolism into pyruvate and further formation of acetyl-CoA, precursor for palmitic acid synthesis, in the reaction catalyzed by pyruvate dehydrogenase complex (PDC), and (2) the reductive route, involving incorporation into TCA cycle in the form of α-ketoglutarate which in the reaction of reductive carboxylation, catalyzed by isocitrate dehydrogenase (IDH), is metabolized into citrate and further utilized for lipogenesis [46].

Redox homeostasis, predominantly regulated by NADPH and reduced glutathione (GSH) molecules that neutralize reactive oxygen species (ROS), is critical for proper cell function [47]. The glutamate metabolized from Gln is a substrate in the first step of GSH synthesis, where γ-glutamylcysteine is obtained from glutamate and cysteine in a reaction catalyzed by cysteine ligase [48].

Additionally, glutamate generated from Gln can be utilized as a substrate for the synthesis of other NEAAs, including alanine, aspartate, proline, and serine, as well as γ-aminobutyric acid (GABA) [49].

The NEAA asparagine is the only one generated from Gln and not glutamate in the reaction catalyzed by the asparagine synthetase (ASNS) [50].

## 3. Dysregulated Glutamine Metabolism in Cancer

In cancer cells, same as in normal cells, Gln contributes to multiple different processes. However, to meet the energy and biomass production requirements, Gln metabolism is accelerated in cancer cells in comparison to normal, non-proliferative cells, and thus the demand is higher. One of the first observations showing enhanced glutamine metabolism in cancer was described by Roberts et al., who found identyfied levels of glutamine in tumors in comparison with normal tissue [51]. This observation was followed by multiple other studies in which dysregulated glutamine metabolism, in the context of various cancers, was described [52,53,54,55,56,57,58]. The summary of the Gln-related dysregulations in the context of involved molecules and cancer type is provided in Table 1.

Overall, in cancer cells, there is a significant demand for Gln, which can be either synthesized or obtained from the bloodstream. The enhanced transport of Gln from the system into the cancer cell can be linked with the observed decrease in the plasma level of Gln in cancer patients across various cancer types [54,59,60]. In fact, enhanced Gln uptake by tumors was suggested to be utilized for tumor imaging where glucose-based positron emission tomography (PET) imaging with 18F-fluorodeoxyglucose (18F-FDG) is inaccurate due to the high background, especially in tissue with high glucose demand, e.g., brain [61]. An investigational PET radiotracer, the 18F-(2S,4R)-4-fluoroglutamine (18F-FGln), was proven as a sensitive strategy to monitor Gln transport and metabolism in human malignancies [62].

**Table 1 cancers-14-00553-t001:** An overview of metabolic fate of glutamine across different cancers.

Pathway	Involved Molecules	Cancer Type	Study Type	References
Increased glutamine transport	**SLC1A5**	Lung cancer	Clinical and in vitro	[63]
		Breast cancer (TNBC)	In vitro and in vivo	[64]
		Head and neck cancer	In vitro and in vivo	[65]
		Colorectal cancer	In vitro and in vivo	[66,67]
	**SLC6A14**	Pancreatic cancer	Clinical, in vitro, and in vivo	[68]
	**SLC38A5**	Breast cancer (TNBC)	Clinical, in vitro, and in vivo	[69]
		Pancreatic cancer	Clinical and in vivo	[70]
Increased glutamine/arginine transport	**SLC6A14**	Cervical cancer	Clinical	[71]
		Colorectal cancer	Clinical	[72]
		Breast cancer (ER^+^)	In vitro and in vivo	[73]
Increased glutamine efflux	**SLC7A5**	Colorectal cancer (K-Ras mutation)	In vivo	[74]
Increased glutaminolysis	**GLS1**	Breast cancer	Clinical, in vitro, and in vivo	[75,76,77]
		Prostate cancer	Clinical and in vitro	[78,79,80]
		Colorectal cancer	Clinical, in vitro, and in vivo	[81]
		Lung cancer	Clinical, in vitro, and in vivo	[82]
Increased glutaminolysis	**GLS2**	Pancreatic cancer	In vivo	[83]
Controls glutamine metabolism and ROS level	**GLS2 ***	Hepatocellular cancer	In vitro	[84,85]
Glutamine contributes to antioxidative capacity of cancer cell	**GCL**	Breast cancer	In vitro and in vivo	[86]
		Lung cancer	In vitro and in vivo	[87]
		Liver cancer	In vivo	[88]
	**GDH1**	Lung cancer	In vitro and in vivo	[89]
		Breast cancer	In vitro and in vivo	[90]
	**GOT1/GOT2**	Pancreatic cancer	In vitro and in vivo	[91]
	**GOT2**	Pancreatic cancer	In vitro	[92]
Glutamine contributes to citrate and lipid synthesis through reductive carboxylation (RC) of α-ketoglutarate (αKG) as well as contributing to aspartate and pyrimidine synthesis	**IDH2**	Renal cell carcinoma deficient in the von Hippel–Lindau (VHL) tumor suppressor gene	In vitro and in vivo	[93]
		Renal cell carcinoma and glioblastoma	In vitro	[94]
Glutamine oxidation maintains TCA cycle	**GDH1**	Lung cancer	In vitro and in vivo	[95]
		Glioblastoma	In vitro	[96]
Glutamine contributes to *de novo* nucleotide synthesis	**GMPS**	Prostate cancer	Clinical and in vitro	[97]
	**GLS1**, **PPAT**, and their ratio **PPAT/GLS1**	Lung cancer/potential role in other cancers	In vitro and in vivo	[98]
	**PPAT** and **PAICS**	Lung cancer	Clinical, in vitro, and in ovo	[99]
	NA	Breast cancer with SIRT3 loss	In vitro and in vivo	[100]
Glutamine contributes to *de novo* asparagine synthesis	**ASNS**	Different cancer cell lines	In vitro	[101]
		Lung cancer	Clinical and in vitro	[102]
Glutamine synthesis	**GLUL**	Pancreatic cancer	Clinical, in vitro, and in vivo	[103,104]
		Glioblastoma	Clinical, in vitro, and in vivo	[105]

The key metabolic enzymes contributing to Gln metabolism: SLC1A5, neutral amino acid transporter belonging to the solute carrier (SLC) family 1 member 5; SLC6A14, neutral and basic amino acid transporter belonging to SLC family 6 member 14; SLC38A5, neutral amino acid transporter belonging to SLC family 38 member 5; SLC7A5, essential amino acid transporter, neutral amino acid antiporter belonging to SLC family 7 member 5; GLS1, glutaminase (characterized as kidney (also known as brain)-type); GLS2, glutaminase (characterized as liver-type); GCL, glutamate cysteine ligase; GDH1, glutamate dehydrogenase 1; GOT1, glutamate oxaloacetate transaminase 1 (cytosolic); GOT2, glutamate oxaloacetate transaminase 2 (mitochondrial); IDH2, isocitrate dehydrogenase 2 (mitochondrial); GMPS, guanosine monophosphate synthetase; PPAT, phosphoribosyl pyrophosphate amidotransferase; PAICS, phosphoribosylaminoimidazole carboxylase and phosphoribosylaminoimidazole succinocarboxamide synthetase; ASNS, asparagine synthetase; GLUL, glutamate ammonia ligase (also known as glutamine synthase). Other abbreviations: ROS, reactive oxygen species; TCA, tricarboxylic acid; TNBC, triple-negative breast cancer; ER^+^, estrogen-receptor-positive; K-Ras, Kirsten rat sarcoma virus. SIRT3, sirtuin 3 (mitochondrial). “*” reflects decreased expression of GLS2 supporting growth of hepatocellular cancer.

Many transporters enabling Gln transport but also antiport were found to be altered across different cancers [31]. The role of SLC1A5 in Gln transport mediation was shown in various lung cancers [63]; breast cancers, including triple-negative breast cancer [64]; head and neck cancer [65]; and colorectal cancer [66,67]. A novel variant of SLC1A5 residing in the inner mitochondrial membrane was recently reported as a key contributor to mitochondrial Gln metabolism and metabolic reprogramming in pancreatic cancer, further pinpointing the importance of Gln in mitochondrial metabolism [106]. The interplay between SLC1A5 and SLC7A5, belonging to the SLC7 family of glutamine antiporters, was previously suggested, pointing towards the enhanced entry of essential amino acids via SLC7A5, activated by Gln flux through SLC1A5, as a strategy contributing to cancer progression [66]. In agreement, the critical role of SLC7A5, exporting glutamine in exchange for essential amino acids to meet cellular demands, in colorectal cancer progression and metastasis was recently reported [74]. However, a study by Cormerais et al. showed the capacity of SLC1A5 as an independent transporter promoting lung tumor growth [66], further suggesting that the tumor-specific landscape of Gln transporters adjusts to meet the metabolic demand.

There are also other Gln transporters, including SLC6A14 and SLC38A5, reported to be upregulated in various tumor types (for SLC6A14: cervical cancer [71], colorectal cancer [72], estrogen-receptor-positive breast cancer [73], and pancreatic cancer [68]; for SLC38A5: TNBC [69] and pancreatic cancer [70]) and contributing to the overall Gln pool, which further highlights the importance of this nutrient in cancer cell metabolism. Furthermore, enhanced expression of SLC38A5 was shown to promote glutamine dependence and oxidative stress resistance in breast cancer [107]. The enhanced expression of different Gln transporters could be multifactorial and might depend on the cancer type, stage, and site, as well as the overall metabolic balance and interplay between cancer and its microenvironment.

The enhanced Gln catabolism in cancer cells was shown to be streamed predominantly to supply TCA cycle, glutathione metabolism [9], and synthesis of lipids [56] and NEAAs [108]. The enzyme GLS1, enabling further processing of Gln and its incorporation into various pathways, was reported as significantly upregulated in a broad spectrum of cancers, including breast [75,76,77], prostate [78,79,80], colorectal [81], and lung [82] cancers, and was shown to be crucial for cancer cell fitness. Interestingly, the other glutaminase isoform, namely GLS2, was observed to be suppressed during malignant transformation [109]. Moreover, GLS2 overexpression greatly reduced tumor cell colony formation and inhibited cancer cell proliferation, suggesting its antitumoral function [84,85]. However, GLS2 expression was shown to strongly depend on the microenvironment as enhanced GLS2 expression was reported under hypoxia in pancreatic cancer [83]. Those studies further highlight the importance of environmental factors contributing to metabolic alterations and adding to the complexity of the network.

The glutamate generated under enhanced GLS1 was shown to serve as a substrate for glutathione synthesis to protect cancer cells from oxidative stress, a feature important for their acquired drug resistance [86]. The glutamate cysteine ligase, catalyzing the first and rate-limiting reaction in glutathione synthesis, where glutamate serves as a substrate, was reported as upregulated in cancer and associated with drug resistance [87,110]. Interestingly, in liver cancer, depletion of glutamate cysteine ligase was associated with decreased glutathione level and higher sensitivity to oxidative stress [88], further highlighting the importance of redox balance in cancer.

Nevertheless, glutathione metabolism is not the only pathway supported by glutamate, as its enhanced contribution to the TCA cycle in the form of α-ketoglutarate was found in malignant cells [9,95]. The metabolism of glutamate into α-ketoglutarate is catalyzed by glutamate dehydrogenase 1 (GDH1), glutamate pyruvate transaminase 2 (GPT2), or glutamate oxaloacetate transaminase 1 and 2 (GOT1 (cytoplasmic) and GOT2 (mitochondrial)), which were al previously reported to be overexpressed in cancer [89,90,91,92]. The activity of enzymes catalyzing glutamate metabolism and the metabolic route of the produced α-ketoglutarate depends on cancer type and metabolic condition of the cell, e.g., nutritional status, and is optimized towards proliferative advantages to cancer cells. For instance, the overexpression of GDH1, observed in breast and lung cancer, associated with tumor progression [89,111], can be related to redox homeostasis beyond the role of feeding the TCA cycle intermediates. Jin et al. suggested that α-ketoglutarate formation controls the level of fumarate, which binds and activates the ROS-scavenging enzyme GPx1 [89]. However, different metabolic dependencies were found in pancreatic cancer, in which two subsequent reactions occur: (1) noncanonical pathway, metabolizing glutamate to aspartate and α-ketoglutarate through GOT2, followed by (2) metabolism of aspartate to oxaloacetate by GOT1 [91,112]. Interestingly, the activation of these reactions was shown to increase the NADPH/NADP+ ratio, thereby maintaining ROS balance, and was indicated as a key pathway for coping with oxidative stress in pancreatic cancer [91]. In breast cancer, reversible transamination catalyzed by GPT2 was shown to be channeled towards glutamate rather than α-ketoglutarate and was linked with the enhanced need for building block production in those cells [90]. On the other hand, the metabolic rerouting towards glutamate generation can also support the maintenance of the cellular redox balance. However, not all cancers utilize the given pathway in a similar manner. For instance, in colon cancer cells, GPT2 was shown to enhance α-ketoglutarate synthesis and incorporation as a Gln-delivered carbon source for the TCA cycle under the enhanced Warburg effect [113]. The diverse strategies of deploying enzymatic machinery further point towards dynamic metabolic adjustments optimized for cancer cell demands.

Upon incorporation into the TCA cycle, α-ketoglutarate can undergo different routes while adjusting to the nutritional status, proliferation rate, and environmental stimuli, which all impact the directionality (oxidative vs. reductive) of the TCA cycle [58]. Citrate, required for the formation of lipid structures to cope with cell proliferation, might be deficient in cancer cells exhibiting the Warburg effect and can be supported by reductive carboxylation of α-ketoglutarate [56]. In this process, mitochondrial isocitrate dehydrogenase (IDH2) catalyzes the metabolism of α-ketoglutarate into isocitrate, which can then be isomerized to citrate [94]. The increase in the α-ketoglutarate/citrate ratio signals reductive carboxylation [114], which was shown to also be upregulated under hypoxia [94]. Under hypoxia, along with the enhanced reductive carboxylation, an elevated level of 2-hydroxyglutarate (2HG) was detected [94]. Interestingly, the 2HG was recognized as an epigenetic modifier and potent oncometabolite [115,116].

Although reductive carboxylation is frequently deployed by cancer cells, the oxidative pathway can also be activated in response to various stimuli to resist metabolic stress [95,96]. For instance, Yang et al. showed that inhibition of the mitochondrial pyruvate carrier (MPC), functioning as a pyruvate supply, contributes to GDH activation further rerouting α-ketoglutarate to generate both oxaloacetate and acetyl-CoA, enabling proper TCA cycle function [95]. Similar observations were made in glioblastoma cells, where under impaired glucose metabolism, glutamine was shown to supply the TCA cycle and production of oxaloacetate [96].

Yet, the latest study by Kodama et al. suggested that a shift from the TCA cycle towards nitrogen metabolism is a rather crucial feature of malignant progression in cancer [98]. The authors showed that an increase in phosphoribosyl pyrophosphate amidotransferase (PPAT) and PPAT/GLS ratio is required to trigger nitrogen shift towards nucleotide biosynthesis in lung cancer and could be considered as a universal phenomenon in other cancer types apart from colorectal cancer [98]. Moreover, the authors further suggested, contrary to the previous observations, that GLS1 possesses antitumor effect rather than protumoral activity, which would call into question all previous reports showing, for instance, involvement of Gln in cancer cell redox homeostasis [117] or lipogenesis [56] and aspartate [108] metabolism. Thus, it would be interesting to investigate multiple tumors under various conditions and assess whether the tumor heterogeneity could explain the inconsistency between the recent report and previous studies. It could be hypothesized that a subpopulation of cancer cells would activate the Gln shift towards nucleotide generation whereas the other population of cells would sustain the Gln incorporation into the TCA cycle to ensure either the oxidative or reductive pathway and lipogenesis. In fact, the cancer cell metabolic program depends on various factors modulating cellular adjustments. The metabolic subpopulations of cancer cells inside the tumor, exhibiting different metabolic programs, were shown to exist in symbiosis, contributing to tumor growth [83]. Given that different populations of cancer cells reside in the tumor, it could be reasoned that both metabolic strategies can be utilized in parallel. Moreover, the utilization of Gln to supply nucleotide synthesis in malignant cells was previously described [99,100,118]; however, the actual shift from the TCA cycle and glutathione metabolism was not previously reported.

*De novo* asparagine synthesis, catalyzed by ASNS, requires glutamine, in contrast to other NEAAs, which can be obtained from glutamate [50]. The expression levels of ASNS vary across different cancers and impact the asparagine level. For instance, in acute lymphoblastic leukemia (ALL), in which low ASNS expression was found, further depletion of asparagine achieved by the introduction of asparaginase (hydrolyzing asparagine to aspartic acid and ammonia) was shown to be lethal in ALL cells and used as a treatment strategy [119,120]. However, cancers with elevated ASNS expression showed resistance to asparaginase [119]. Moreover, elevated ASNS level was shown to promote cancer cell proliferation [101] and was linked to drug resistance [119,121,122].

Finally, the high and diverse tumor demands for glutamine might be challenging to meet under steady state in which Gln production is predominantly supported by the muscle, which further suggests a need for activation of Gln synthesis in some cancers. Indeed, enhanced proliferation rate, along with faulty vascularization in, e.g., pancreatic cancer, result in scarce access to the nutrients and activation of other processes such as extracellular protein scavenging [104]. Furthermore, glutamine synthesis, but not glutaminolysis, was found to be upregulated in pancreatic cancer [103,123] and glioblastoma [124]. GS activity was linked with nucleotide synthesis in glioblastoma under glutamine restriction [105], and *de novo* synthesized glutamine was shown to be essential for nitrogen anabolic processes in pancreatic cancer [103,125].

In summary, there are multiple metabolic processes activated in cancer cells that rely on Gln. The way in which Gln metabolism is altered in the tumor is multifactorial and depends on cancer type and site, nutrient access, and molecular signaling.

## 4. Key Components Regulating Glutamine Metabolism

### 4.1. Environmental Factors

The rapid cancer cell proliferation resulting in tumor tissue growth leads to enhanced oxygen demand, which could not be fulfilled due to distanced existing vasculature or defective angiogenesis causing limited oxygen supply and thus hypoxia (low oxygen level in tumor tissue) [126,127]. The hypoxic microenvironment was shown to be implicated in metabolic reprogramming, further contributing to cancer cell progression and aggressiveness [128,129]. The adaptation to hypoxia can be modulated by hypoxia-inducible factors HIF-1α and HIF-2 α, overexpressed by cancer cells in response to low oxygen levels [130], which in turn induce metabolic genes involved in glycolysis and Gln metabolism [131,132]. It was shown that HIF-1α suppresses glucose-dependent anabolic synthesis with a simultaneous increase in lactate production [131]. Hypoxia-driven alterations result in a reduction in pyruvate entering the TCA cycle [133,134], which could further trigger reprogramming in Gln metabolism. Indeed, enhanced Gln transport and upregulation in the expression of Gln transporters were observed under hypoxia [106,135,136]. The expression of SLC1A5 is controlled and upregulated by HIF-2α [106], whereas SLC38A1 is activated by HIF-1α [135] and presents different strategies to modulate the Gln level in the cell.

Yet, the Gln transported into the cell is further utilized to sustain the needs of hypoxic cells in which oxidative pathway changes more towards reductive carboxylation [136,137]. The utilization of glutamine under hypoxia is enhanced, and glutaminolysis catalyzed by increased activation of both GLS1 and GLS2 was shown to be upregulated [81,83]. The activation of GLS1 was found to be critical for colon cancer growth [81], whereas hypoxia-activated GLS2 was found in pancreatic cancer [83]. Further, the Gln-derived carbon is utilized for the formation of α-ketoglutarate, which undergoes reductive carboxylation catalyzed by IDH2 to form isocitrate followed by isomerization to citrate [94]. Noteworthily, this metabolic upregulation was shown to be associated with increased synthesis of 2HG, recognized as an oncometabolite [94,115]. The generated citrate is further transported into the cytoplasm to generate acetyl-CoA, which serves as a source for fatty acid synthesis [138,139]. The utilization of Gln as a lipid source was recognized as critical for the proliferation of cancer cells under hypoxia in multiple studies [56,92,94,132,136,137]. Interestingly, hypoxia was shown to inhibit carnitine palmitoyltransferase 1A (CPT1A) and thus beta-oxidation, which resulted in the accumulation of lipids in form of stored droplets in clear cell renal cell carcinoma [140]. It could be reasoned that cancer cells might store fatty acids generated under reductive carboxylation for their potential future utilization under changed conditions. Given that reductive carboxylation was shown to be necessary for tumor growth under hypoxia, this pathway could be considered as a promising treatment target [132].

Nevertheless, under hypoxia, enhanced oxidative Gln metabolism resulting in ATP production in addition to glutathione synthesis can be maintained in some cancers despite decreased mitochondrial respiration [135,141,142,143]. In a Myc-inducible human Burkitt lymphoma model, oxidative Gln metabolism supporting the TCA cycle was reported under hypoxia [142], further suggesting the potential involvement of the Myc oncogene in determining the fate of glutamine. The differences in TCA directionality under hypoxia could reflect the cellular demands at a given time and further point towards the metabolic plasticity of cancer cells.

The fate of Gln-derived nitrogen under hypoxia was also investigated. Noteworthily, the utilization of Gln-derived carbon for lipogenesis generates byproducts such as nitrogen (ammonia/ammonium) and oxaloacetate that could not be effectively catabolized under limited oxygen [136] and could be even toxic to the cell [144]. Therefore, under the hypoxic condition, the cancer cell activates an alternative pathway in which Gln-derived nitrogen is incorporated into dihydroorotate. The conversion of dihydroorotate to UMP was shown to be hampered and followed by dihydroorotate accumulation and further excretion [136,145]. The observed accumulation of dihydroorotate indicates a cellular strategy to scavenge potentially toxic nitrogen byproducts resulting from enhanced Gln utilization for lipogenesis under hypoxia.

### 4.2. Oncogenes and Tumor Suppressors Control Altered Glutamine Metabolism

The metabolic landscape of the tumor is dictated not only by the environmental factors but also by the oncogenes and tumor suppressors. For instance, tumor suppressors such as retinoblastoma protein (Rb), sirtuin 4 (SIRT4), and p53 were linked with dysregulated Gln metabolism in various cancers [53,84,85,146,147,148].

The depletion of Rb was associated with a specific increase in Gln uptake via upregulated SLC1A5 [53,146] and enhanced glutathione formation [146]. Furthermore, under Rb loss, enhanced glutaminolysis mediated by GLS1 and Gln incorporation into the TCA cycle was found and linked with increased ATP production [53], further suggesting activation of the oxidative pathway. Similarly, loss of SIRT4 was also associated with enhanced Gln transport and glutaminolysis contributing to enhanced proliferation and growth of lung and breast cancers [147,148].

The role of p53 has grown lately beyond the canonical activities regulating cell division, as it has been shown to have a function in cellular metabolism involved in responses to metabolic stresses [149]. For instance, p53 was shown to be activated in response to low Gln level [150], in turn promoting expression of different transporters, including SLC1A3 (glutamate/aspartate transporter) [151] and SLC7A3 (enhancing arginine import) [152]. The enhanced SLC1A3 expression, modulated by p53 in response to low Gln level, was shown to rescue cell viability by maintaining TCA cycle activity and promoting *de novo* synthesis of Gln, glutamate, and nucleotides [151]. The p53-modulated enhancement of different transporters presents strategies enabling cancer cells to facilitate adaptation to Gln deprivation.

The role of oncogenes were identified as critical for controlling Gln uptake and metabolism and the contribution of Gln metabolism to malignant transformation. For instance, oncogene c-Myc coordinates the expression of genes required for Gln catabolism by reprogramming the mitochondrial metabolism towards addiction to glutaminolysis [153]. These changes were linked with suppression of glucose incorporation into the TCA cycle with simultaneous enhancement of GLS1-mediated glutaminolysis [52] and Gln streaming to the TCA cycle [153]. Interestingly, the contribution of c-Myc to Gln synthesis was also reported, suggesting the involvement of c-Myc in both anabolic and catabolic pathways [154] potentially to maximize utilization of Gln by cancer cells. Similarly to the phenotype observed in tumors with Rb loss [53], c-Myc overexpression was also likened with enhanced Gln transport via SLC1A5 [153].

Another oncogene, Kirsten rat sarcoma (K-Ras), was also identified as a modulator of metabolism contributing to malignant transformation and cancer progression [75,119]. K-Ras was found to promote glutaminolysis in a GLS1-dependent manner, and the cancer cell survival was shown to be Gln-dependent [155] Moreover, the K-Ras-transformed cells efficiently utilized both Gln-delivered carbon and nitrogen for the synthesis of building blocks, including amino acids and nucleotides, and for glutathione generation [156]. Such an efficient metabolic program enables sustaining growth on one hand and protecting from oxidative stress on the other hand. Interestingly, in K-Ras pancreatic cancer, noncanonical Gln utilization was reported in which glutamine-derived aspartate is converted into oxaloacetate by GOT1 [91]. This non-canonical pathway serves as a key contributor to generating oxidative capacity of the cell by increasing the NADPH/NADP+ ratio in the series of reactions in which oxaloacetate is converted into pyruvate [91]. Moreover, the cell cycle arrest caused by suppression of glutamine utilization in K-Ras-driven cancer cells was reversed by aspartate [157], further underscoring the significance of this noncanonical pathway.

In summary, both environmental factors and molecular signaling pathways contribute to dysregulated Gln metabolism in cancer cells.

## 5. Combined Treatment Strategies Targeting Altered Glutamine Metabolism Offer a Promising Cancer Treatment

The extensive research focused on Gln metabolism in health and disease resulted in the identification of potential treatment targets, which were further probed in preclinical and clinical studies. Overall, modulation of Gln metabolism results in inhibition of cancer cell proliferation; however, it is frequently insufficient to induce cancer cell death, potentially due to activation of compensating pathways. Here, we provide an overview of preclinical and clinical studies in which Gln metabolism was targeted, and we further focus on cancer cell responses to such a treatment, as those can serve as a roadmap for rational designing of combined treatment strategies.

### 5.1. Targeting GLS1 along with Chemotherapy as a Promising Treatment Strategy

The enhanced glutaminolysis orchestrated by GLS1, observed to be essential for multiple cancers [57,117,158,159], was probed as a treatment target with different inhibitors such as bis-2-(5-phenylacetamido-1,2,4-thiadiazol-2-yl)ethyl sulfide (BPTES) [160]; bromo-benzophenanthridinone compound 968 (C.968) [161]; and CB-839, also known as telaglenastat, a product of company Calithera [76]. The inhibition of glutaminolysis was shown to suppress the growth of multiple different cancer cells, including breast [76], renal [93], and liver [162], in vitro and in vivo and was successful in hampering metastatic progression in osteosarcoma [158]. Although GLS1 inhibition leads to suppression of cancer cell proliferation, it is frequently insufficient to trigger cancer cell death, as was shown by us and others, independently of the GLS1 inhibitor used [12,13,93,163,164]. Thus, it can be reasoned that the inactivation of a single metabolic pathway is insufficient to trigger cancer cell death due to potential metabolic compensations. This is in agreement with outcomes from other monotherapies, further suggesting a need for combined treatment strategies [165]. Importantly, we found that GLS1 inhibition resulted in activation of lipid catabolism along with autophagy as a cancer survival mechanism [12], which was confirmed by others [13]. In fact, activated autophagy [166,167] and accelerated lipid catabolism [168] were recognized as pathways supporting cancer cells under nutritional challenges and thus should be considered for cotreatment strategies. Our in vitro experiments showed that simultaneous inhibition of GLS1 along with lipid catabolism or autophagy targeted with trimetazidine and chloroquine, respectively, could be considered as a potential treatment strategy [12]. In fact, this treatment option could be explored given that both trimetazidine and chloroquine are used in clinic. Interestingly, the latest study suggested that metformin, a first-line antidiabetic drug, possesses the ability of GLS1 inhibition [169,170]. However, in contrast to a study in which GLS1 inhibitors were used, metformin treatment resulted in autophagy suppression and apoptosis activation [170]. The differences in cancer cell responses could be explained by the fact that metformin is not a GLS1-specific inhibitor and impacts other pathways relevant for cancer cell growth [171].

Currently, there are a total of 20 registered clinical trials investigating the potential of telaglenastat (CB-839) as an anticancer drug, out of which 8 have been completed (Table 2).

Overall, telaglenastat (CB-839) showed safety and was well tolerated in patients with leukemia and other hematological cancers, as well as those with solid tumors (including clear cell renal cell carcinoma, melanoma, non-small-cell lung cancer, and triple-negative breast cancer) [172,173,174,175]. The telaglenastat treatment was proved to inhibit GLS1 in the patients [175], which is in agreement with the initial preclinical study [76]. However, the observed progression-free survival was moderate [177], and a clinical trial with renal cell carcinoma patients did not achieve the primary endpoint [176]. Noteworthily, in almost all clinical trials, telaglenastat was combined either with chemotherapy or immunotherapy and such a treatment regimen was given to heavily pretreated patients, which could reflect on the trial outcome. Further, the moderate responses to the treatment could be explained by the potential compensation mechanisms activated by the tumor. Thus, in-depth analysis of molecular alterations in response to treatment, preferentially on the metabolic level, should be considered. Moreover, given the complexity of metabolic dependences in the tumor and the symbiosis between the cancer cell populations in the tumor [83], markers enabling better prediction of the responses would be required to guide the selection of treatment strategies. Noteworthily, GLS1 inhibition was shown to lower the level of glutathione, suggesting its impact on the antioxidative capacity of cancer cells and their survival under the treatment [12,76]. Hence, drugs triggering oxidative stress, e.g., doxorubicin, could be considered for combination with telaglenastat (CB-839), given cancer cell dependency on glutathione under doxorubicin treatment [178]. Nevertheless, the preclinical data should be carefully analyzed as doxorubicin-induced site effects related to oxidative stress could strongly affect patient wellbeing [179].

### 5.2. Targeting Antioxidative Capacity of the Cell by Modulating Glutamine Metabolism as a Potential Treatment Option

The antioxidative capacity of a cancer cell supports its fitness and survival [180]. Therefore, targets such as GOT1, which is a contributor to the antioxidative capacity of pancreatic ductal adenocarcinoma (PDA) [91] but is dispensable in nonmalignant cells, could be considered as an attractive treatment option. GOT1 is required for proliferation and tumor growth, and its inhibition leads to attenuated cancer cell progression through cell cycle blockage [181]. Thus, GOT1 could be considered as a promising and cancer-specific treatment target. Recently, aspulvinone O was reported as a natural inhibitor of GOT1 sensitizing PDA cells to oxidative stress and resulting in growth suppression in vitro and in vivo [182]. Furthermore, a covalent small molecule inhibitor (PF-04859989) of GOT1 was recently tested and showed selective growth inhibition of PDA cell lines [183]. Nevertheless, Kremer et al. showed diverse cell sensitivity to GOT1 inhibition further suggesting activation of cancer cell survival mechanisms in response to the treatment [181], further highlighting the need for taking into account metabolic plasticity. Chronic GOT1 suppression resulted in enhanced cystine import, glutathione synthesis, and lipid antioxidant machinery along with activation of glutathione peroxidase 4 (GPX4), which enabled cancer cells to cope with oxidative stress and survive the treatment [181]. The simultaneous inhibition of GOT1 and any of the pathways activated in response to this inhibition resulted in ferroptosis [181], a form of cell death [184], further suggesting a potential treatment strategy for PDA tumors.

Yet another example of an enzyme contributing to the oxidative capacity of cancer cells is GDH1, which could serve as a potential treatment target [89]. Inhibition of GDH1, either by shRNA or small molecule inhibitor R162, was shown to disrupt redox homeostasis in cancer cells and inhibit their proliferation and tumor growth in vitro and in vivo, causing minimal toxicity [89]. Another study by Jin et al. suggested that GDH1 plays an important role during lung cancer metastasis by contributing to α-ketoglutarate synthesis which in turn modulates the 5′ AMP-activated protein kinase (AMPK) signaling pathway [185]. Our recent study showed an increased level of α-ketoglutarate in breast cancer cell lines with high metastatic potential, further suggesting the importance of this pathway in cancer progression [186]. Targeting GDH1 with R162 attenuated tumor metastasis in a patient-derived xenograft model, offering a potential treatment for lung cancer patients [185]. GDH1 could be also targeted with epigallocatechin gallate (EGCG), a flavonoid isolated from green tea [187], which was shown to inhibit lung cancer cell proliferation and invasion in vitro [188]. Importantly, the redox balance and α-ketoglutarate level disrupted by R162 can be further restored by other metabolic pathways, which should be taken under consideration in future studies targeting GDH1.

### 5.3. Starving Cancer Cells of Glutamine as a Cotreatment Strategy

The dysregulated Gln metabolism could also be managed by controlling the amount of Gln entering the cell. This can be achieved by targeting SLC1A5, which was shown as a key Gln transporter contributing to cancer cell proliferation [63,64,65,66,106]. Alternatively, Gln antagonists, by inhibiting Gln-utilizing enzymes, limit the performance of Gln-dependent pathways and thus impact cancer function [189,190,191,192,193].

The inhibition of SLC1A5 with the small molecule L-γ-glutamyl-p-nitroanilide (GPNA) leads to glutamine starvation in lung cancer cells, which further results in oxidative stress-mediated autophagy and apoptosis [194]. Given that autophagy is considered as a survival pathway activated by the cancer cell under nutritional stress [167], its activation under inhibited Gln influx should be carefully monitored. Another pharmacological blockade of glutamine flux was recently reported by Schulte et al., who described 2-amino-4-bis(aryloxybenzyl)aminobutanoic acids (V-9302) as selective and potent competitive small molecule antagonists targeting SLC1A5 [195]. The SLC1A5 inhibition resulted in attenuated colon cancer cell growth and proliferation, increased cell death, and increased oxidative stress in vitro and in vivo [195]. Similarly, a study conducted in head and neck squamous cell carcinoma (HNSCC) targeting overexpressed SLC1A5 with V-9302 reported the suppression of the intracellular glutamine level and downstream Gln metabolism, which in turn inhibited cancer growth and proliferation in vitro and in vivo [65]. Furthermore, the authors showed that inhibition of SLC1A5 sensitized HNSCC to cetuximab, thus suggesting optimization of treatment strategy by using metabolic modulation [65]. Interestingly, SLC1A5 was recently recognized as a potential target for leukemia [196], which could be probed with V-9302. Given the sensitized impact of SLC1A5 inhibition on chemotherapy [65], a similar outcome might be expected in leukemia. Nevertheless, this strategy should be carefully investigated given the potentially severe adverse impact of limited Gln access on normal hematopoiesis in patients. Noteworthily, the decreased Gln level under SLC1A5 inhibition could be compensated for by either activation of alternative Gln transporters or Gln synthesis. In fact, enhanced sensitivity to the inhibition of Gln transporters was linked with reduced plasticity associated with the capability of activation of the alternative pathway [197].

Given that Gln can be generated by the cell, Gln antagonists could be considered as an alternative treatment option to molecules inhibiting Gln transporters. Indeed, Gln antagonist 6-diazo-5-oxo-l-norleucine (DON), identified in 1956 [139], was shown to impact Gln-dependent pathways, including synthesis of nucleotides and α-ketoglutarate [190,198], and thus cancer cell growth in vitro and in vivo [139,189,193,198]. Nevertheless, the initially promising clinical development of DON [199,200] was hampered due to its dose-limiting toxicity related to gastrointestinal (GI) mucosa [193]. The latest development aiming to optimize DON to the form of a prodrug circulating in the plasma and activated in the tumor resulted in the successful finding of (DON-based) Gln antagonist prodrugs that provide selective tumor exposure [191,192,201]. A recent study deploying a DON-based prodrug, the compound JHU083, for glutamine blockade in tumor-bearing mice reported the suppression of oxidative and glycolytic metabolism in cancer cells and a significant decrease in tumor volume [191]. Moreover, another DON-based prodrug called sirpiglenastat (DRP-104), a leading candidate of Dracen company, is currently being probed as a single agent and in combination with immunotherapy (atezolizumab) in a clinical trial in patients with advanced solid tumors [202]. Given that sirpiglenastat can simultaneously hamper multiple Gln-dependent pathways, its overall performance in clinical trials might be superior to that of molecules targeting only one enzyme (e.g., telaglenastat). Although the performance of sirpiglenastat as a single agent might be limited due to the plasticity of cancer cells, combined treatment strategies deploying currently suggested immunotherapy could serve as an attractive option for cancer patients.

In summary, the dysregulated Gln metabolism has become an attractive treatment target. Although there are many strategies involving the modulation of Gln metabolism that were proven efficient in preclinical testing, so far, only two molecules, telaglenastat (CB-839), targeting GLS1, and sirpiglenastat (DRP-104), targeting multiple Gln-dependent enzymes, have reached clinical trials. Further effort toward understanding cancer survival mechanisms under the treatment would be needed to optimize combined treatment options for identified targets.

## 6. Potential Pitfalls and Future Perspectives

Although dysregulated Gln metabolism is an attractive treatment target, which could potentially be applied for different cancers, further optimization of current options is required to achieve better clinical outcomes. Aspects concerning tumor heterogeneity and metabolic plasticity, which both contribute to cancer cell resistance, are critical roadblocks that have to be addressed.

Currently, tumor metabolic status and metabolic dysregulations of cancer cell populations in the tumor and their responses to the stimuli are not considered while designing treatments. However, these factors may be critical given the interplay between different components of the tumor and their metabolic symbiosis [83]. A measurement enabling in-depth characterization of the tumor metabolic processes, ideally in a dynamic manner, would be required for optimized treatment. In fact, metabolomics, which is a strategy for monitoring small molecule composition, offers such a possibility [203]. We used metabolomics to identify cancer compensation mechanisms in response to glutaminolysis inhibition [12] and to identify metabolic pathways activated as a tumor response to doxorubicin treatment [178]. Although such an approach is informative and can result in the identification of optimized treatment strategies, it is not ideal as it provides only a snapshot and not a continuous measurement of the process. Thus, noninvasive approaches enabling continuous and real-time monitoring of metabolic process, which could be incorporated inside the tumors, could be a potential step forward. In fact, nanowire sensors enabling protein and DNA monitoring were recently introduced [204]. Therefore, sensors with the capacity for dynamic metabolic monitoring of the tumors could be envisioned. Such an approach would support the optimization and adjustment of treatment strategies in a dynamic manner.

In conclusion, treatment strategies targeting Gln metabolism constitute an attractive and promising option for cancer patients. However, due to the metabolic plasticity of the tumor, alternative pathways can be activated to overcome the treatment. Thus, identification of the pathways activated to support cancer cell survival under the treatment is critical for further development and optimization of any potential therapy. Given the complexity of the metabolic network, a combined treatment directed towards cancer metabolism could be more successful than monotherapies. Finally, a device enabling continuous monitoring of metabolic processes inside the tumor could support treatment decision making and be a step forward for cancer precision medicine.

## Figures and Tables

**Table 2 cancers-14-00553-t002:** Overview of completed clinical trials testing telaglenastat (CB-839) in various cancer patients.

Cancer Type	Treatment	Outcome	Reference
**Triple-negative breast cancer (TNBC)**	In combination with paclitaxel (chemotherapeutic agent targeting microtubules)	In heavily pretreated patients with previous taxane exposure, the treatment demonstrated clinical activity and was well tolerated.	[172]
**Clear cell renal cell carcinoma (ccRCC), melanoma; non-small-cell lung cancer (NSCLC)**	In combination with nivolumab (immunotherapy medication targeting programmed cell death (PD-1) receptor)	CB-839 was well tolerated when combined with nivolumab in melanoma, ccRCC, and NSCLC patients.	[173]
**Solid tumors with K-Ras mutation**	In combination with palbociclib (kinase inhibitor targeting cyclin-dependent kinases CDK4 and CDK6)	NA	NA
**Solid tumors**	As a single agent and in combination with standard chemotherapy	Acceptable safety profile under continuous CB-839 administration. Treatment resulted in glutaminase inhibition and clinical activity.	[174]
**Hematological tumors**	As a single agent or in combination with pomalidomide (immunomodulatory agent), dexamethasone (glucocorticoid), or pomalidomide and dexamethasone	CB-839 administration was well tolerated and resulted in GLS inhibition in blood platelets and in tumors. Observed reductions in marrow and peripheral blast counts suggested clinical relevance.	[175]
**Renal cell carcinoma (RCC)**	In combination with cabozantinib (tyrosine kinase inhibitor)	Did not achieve primary endpoint.	[176]
**ccRCC**	In combination with everolimus (mammalian target of rapamycin (mTOR) kinase inhibitor)	In combination with everolimus, CB-839 demonstrated a tolerable safety profile. Modest (3.8 months from 1.9 months) progression-free survival was observed.	[177]
**Leukemia**	CB-839 as a single agent or in combination with azacitidine (chemotherapeutic agent, antimetabolite)	CB-839 was well tolerated in advanced leukemia and resulted in GLS inhibition in platelets and PBMCs. Two patients achieved significant reductions in blast counts.	[175]

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
