# Peer review of "Advancing Cancer Treatment by Targeting Glutamine Metabolism—A Roadmap"

_cancers, 2022, doi:10.3390/cancers14030553_

Round 1
Reviewer 1 Report
In this review, Dr. Suhre, an expert in Biophysics and metabolomics biology, provides a comprehensive description of how combined treatment strategies targeting glutamine metabolism along with other pathways as potential treatment options. It is a very important work in the field of Advancing Cancer Treatment by Targeting Glutamine Metabolism and provides valuable information to the readers. The contents are comprehensive and cover the latest findings, and no major revisions are considered necessary. The following points could use some improvement.
- Please discuss the latest findings from Leone et.al., (Science 2019) on the Glutamine blockade that induces divergent metabolic programs to overcome tumor immune evasion. Importantly, discuss how the Glutamine antagonism exposes a previously undefined difference in metabolic plasticity between cancer cells and effector T cells that can be exploited as a “metabolic checkpoint” for tumor immunotherapy.
- It is interesting to take into the consideration and discus about the novel glutamine antagonist DRP-104 (Sirpiglenastat) Dracen Pharmaceutical’s, Granted U.S. FDA fast track designation for the treatment as Single Agent and in Combination with Atezolizumab in Patients with Advanced Solid Tumors.
- The main focus of this review, the perspective combined treatment strategies targeting glutamine metabolism along with other pathways as potential treatment options. The comparison of different cancer types is interesting, but looking at table 2, there are many differences in the treatment and outcome. Does the author think that some common mechanism is used in the developmental stage where glutamine accumulation formed? Or are they completely different processes? I propose to add a discussion of the similarities in particular and if possible, include them in Table 2.
- It is interesting to discuss what the author thinks about the combination therapy of Glutamine transporter (V-9302) & glutamine antagonist in Cancer cure.
Author Response
We appreciate the comments and suggestions raised by the reviewer. Please find attached our point-by-point responses.

Reviewer 2 Report
I have reviewed the manuscript entitled” Advancing Cancer Treatment by Targeting Glutamine Metabolism – A Roadmap" by Suhre and Halama. In this study, authors described that the importance of the glutamine metabolism is one of the metabolic pathways that can be successfully targeted in cancer treatment. Finally, the authors put into perspective the strategies that implement a combination treatment to target glutamine metabolism alongside other molecular or metabolic pathways and discuss their potential for clinical applications.
The description of the legend in table 1 is poor. Some grammar errors also transpired in the text, a moderate English changes required.
In paragraph 5.1 you can consider this reference: DOI 10.3390/cells8010049 Metformin impairs glutamine metabolism and autophagy in tumour cells.
References are not sufficient to cover all previous studies related to this work, with particular reference to the hypoxia pathway and glutamine metabolism.
Author Response

(The authors gave the same response as above.)
